# Wolves and dogs fail to form reputations of humans after indirect and direct experience in a food-giving situation

**Hoi-Lam Jim**[ORCID]\*, **Marina Plohovich, Sarah Marshall-Pescini, Friederike Range**\*

Domestication Lab, Konrad Lorenz Institute of Ethology, Department of Interdisciplinary Life Sciences, University of Veterinary Medicine Vienna, Vienna, Austria

\* hoi-lam.jim@outlook.com (HLJ); friederike.range@vetmeduni.ac.at (FR)

## Abstract

Reputation is a key component in social interactions of group-living animals and appears to play a role in the establishment of cooperation. Animals can form reputations of individuals by directly interacting with them or by observing them interact with a third party, i.e., eavesdropping. Previous research has focused on whether dogs (*Canis lupus familiaris*) can eavesdrop on humans because of their ability to cooperate with humans, however the results are mixed and if they can eavesdrop, it is unknown whether this ability evolved during the domestication process or whether it was inherited from their ancestor, wolves (*Canis lupus*). Our study investigated whether equally hand-raised, pack-living dogs and wolves can form reputations of humans in a food-giving situation through indirect and/or direct experience. The experimental procedure comprised three parts: baseline (to test whether the subject preferred a person prior to the experiment), observation and testing. In the observation phase, the subject observed two humans interact with a dog demonstrator–one acted generously and fed the dog, and the other acted selfishly and refused to feed the dog. The subject could then choose which person to approach in the test phase. In the following experience phase, the animals interacted directly with the same two humans who behaved either in a generous or selfish manner. Then, they were again given a choice whom to approach. We found that dogs and wolves, at the group level, did not differentiate between a generous or selfish partner after indirect or direct experience, but wolves were more attentive towards the generous person during the observation phase and some dogs and wolves did prefer the generous partner, at least after indirect and direct experience was combined. Our study suggests that reputation formation may be more difficult than expected for animals and we emphasise the importance of context when studying reputation formation in animals.

## Introduction

Cooperation can be defined as two or more individuals working together to achieve a common goal [1]. Group-living animals often cooperate to increase their chance of survival, for

**Data Availability Statement:** All relevant data are within the manuscript and its Supporting Information files.

**Funding:** HLJ and FR were funded by the Austrian Science Fund (FWF): W1262-B29 (www.fwf.ac.at). The funders had no role in study design, data collection and analysis, decision to publish, or preparation of the manuscript.

**Competing interests:** The authors have declared that no competing interests exist.

example, wolves (*Canis lupus*) work together to hunt down large prey [2], meerkats (*Suricata suricatta*) rear pups communally [3] and male bottlenose dolphins (*Tursiops aduncus*) form alliances to monopolise access to females [4]. Cooperation may lead to a mutual benefit to all individuals involved, such as sharing resources, but this is not always the case. For example, some individuals may not be willing to share the resources obtained through cooperation, resulting in some cooperative individuals not receiving their due benefits, or there may be free riders who receive the benefits from cooperation even though they did not help to achieve it. Therefore, it is important for animals to choose 'good' partners with whom to cooperate. Reputation formation is the ability to gain knowledge about how an individual may behave in the future based on how they behaved in the past [5]; consequently, an individual with a reputation of being 'cooperative' may be more likely to gain access to valuable resources and partners, whereas an individual with a reputation of being 'non-cooperative' may be excluded from social interactions with others [6].

Animals can form reputations of individuals by directly interacting with them or by observing them interact with a third party, known as eavesdropping [7]. Direct experience with an individual provides the most accurate prediction of an individual's future behaviour but it may be costly, for example, if an individual is aggressive or dangerous. Thus, eavesdropping is a useful skill to predict the behaviour of others without the costs of direct experience, but it is presumed to be more cognitively demanding than direct reputation formation because it requires individuals to remember and recognise behaviours in third-party interactions [8]. The most common method to investigate reputation formation in animals is for the subject to interact (direct reputation formation) or observe another individual interact (eavesdropping) with two partners with opposing roles, such as a generous/helpful and selfish/unhelpful partner, and then observe with whom the subject chooses to interact afterwards.

A few studies have investigated direct reputation formation using interactions between conspecifics in chimpanzees (*Pan troglodytes*) [9] and coral trout (*Plectropomus leopardus*) [10] and showed that animals learnt to recruit the more effective partner when solving a problem that required collaboration. However, research on eavesdropping has often studied non-human primates and involved the animal observing human-human interactions in a food-giving situation (e.g., [5,8,11]), where one experimenter was 'nice' and gave food to a human beggar, and another experimenter was 'mean' and did not allow the beggar to eat. Russell et al. [5] and Herrmann et al. [11] showed that chimpanzees significantly preferred the nice experimenter over the mean one and Subiaul et al. [8] found that chimpanzees only preferred the nice experimenter after they observed interactions between a human and a conspecific recipient but not when the interaction was between two humans. This suggests that human-animal interactions may potentially be more salient or relevant than human-human interactions, thereby enhancing the animals' performance. Nevertheless, non-human primates' social interactions are exclusively between conspecifics in the wild, so the studies mentioned above involving human partners still lack ecological validity.

Contrariwise, information regarding humans may be highly relevant for domestic animals, who depend on them for survival. Dogs (*Canis lupus familiaris*) may be particularly sensitive towards human social behaviours because they were the first species to be domesticated [12] and, due to their close co-habitation, may have acquired advanced sociocognitive abilities (Emotional Reactivity Hypothesis [13]) or social competence skills that have enabled them to cooperate with humans (Social Competence Hypothesis [14]) as a result of the domestication process.

Studies on direct reputation formation in dogs have shown mixed results. Nitzschner et al. [15] found that dogs spent significantly more time close to the 'nice' human, who was friendly and playful towards the dog, than the 'ignoring' human. In Heberlein et al. [16], dogs led a

cooperative partner more often than a competitive partner to a box containing food, as the cooperative partner rewarded the dog with the food inside the box and the competitive partner took the food out of the box and put it in her pocket. This demonstrates that dogs distinguished between the cooperative and competitive partner based on their direct experience. Moreover, two studies by Carballo et al. [17,18] showed that dogs preferred to approach and gazed more at a generous human than a selfish one in a food-giving situation [17] and when confronted with an unsolvable task [18]. However, other studies have found that dogs could not form reputations of humans after direct experience–Piotti et al. [19] could not demonstrate that dogs formed a reputation of an experimenter based on her skilfulness or the quality of the interaction and McGetrick et al. [20] found that dogs did not prefer a helpful human, who provided them with food by activating a food dispenser, compared to an unhelpful human, who did not provide them with food, which suggests that they did not form reputations of the humans based on their cooperativeness.

Studies on eavesdropping in dogs have also shown mixed results. Several studies using human-human interactions have found that dogs significantly preferred the generous person over the selfish one in a food-giving situation [21,22] and the helpful person over a neutral [23] or an unhelpful person in a helping situation [24]. However, these positive results have been questioned, as the dogs' behaviour could be explained by simpler mechanisms, such as local enhancement [25–27]. These results highlight the importance of designing experiments with adequate controls to elucidate dogs' ability to eavesdrop.

As mentioned above from Subiaul et al.'s findings [8], human-animal interactions may enhance the relevance of the interactions for the observing animal compared to human-human interactions, however far fewer studies on eavesdropping in dogs have been conducted using this setup. Rooney and Bradshaw [28] found that dogs preferred to approach a person who won a tug-of-war game with another dog over a person who lost the game, but Nitzschner et al. [15] found that dogs did not prefer an experimenter who was nice to another dog compared to an experimenter who ignored the dog. Nevertheless, they did find that subjects behaved differently during the nice and ignoring demonstrations, such as vocalising, scratching and jumping up more and looking longer towards the nice experimenter, which suggests that they were attentive to the different types of dog-human interactions but the experimenters' behaviour towards the dog demonstrator may not have been relevant enough for the observer dog to form reputations of them.

As humans artificially selected dogs to have explicitly desired traits, such as the ability to cooperate with us [29], their ability to eavesdrop may be a novel ability that emerged during the domestication process. Conversely, it may have derived from their closest living relatives, wolves, which are highly cooperative with each other. Range and Virányi postulated that dog-human cooperation evolved on the basis of wolves' cooperative skills (Canine Cooperation Hypothesis [30]), and studies conducted at the Wolf Science Center (WSC) have shown that similarly raised and kept wolves and dogs with comparable experience of humans have similar sociocognitive skills–wolves are as skilled as dogs, if not better, at following human gazing cues into distant space and around barriers [31,32] and at following human communicative cues (i.e., looking or pointing) [33], and they do not differ in their capacity to learn from human partners [34]. Even more importantly, wolves and dogs can cooperate with humans [35] and can even recruit human partners in a cooperative string-pulling task [36]. These findings provide support for the Canine Cooperation Hypothesis [30] and it is plausible that wolves and dogs are able to eavesdrop, as it could facilitate cooperation.

Although it is unclear whether dogs can eavesdrop, we tested wolves and dogs at the WSC to disentangle if dogs have this skill, whether it evolved during the domestication process or was inherited from their ancestor. A study similar to Heberlein et al. [16] was conducted at the

WSC–the authors tested dogs' and wolves' ability to 'communicate' with humans, with the cooperative partner giving the animal the food reward and the competitive partner eating the food herself [37]. Then, the subject could use 'showing' behaviours to indicate a food location to the partner. They found that dogs and wolves showed the food location to the partner similarly often and both exhibited more showing behaviours in the presence of the cooperative partner than the competitive one, which indicates that they differentiated between the cooperative and competitive partner based on their direct experience. However, until now, no study has investigated eavesdropping in wolves.

The aim of this study was to test whether dogs and wolves can form direct and/or indirect reputations of humans using human-animal interactions in a food-giving situation. We controlled for local enhancement and other potential confounding variables by counterbalancing the partners' roles and the colour of their clothes, which also helped to make the partners' distinguishing features more salient. Based on previous studies conducted at the WSC that showed similarly raised and kept wolves and dogs have similar sociocognitive skills with humans, which supports the Canine Cooperation Hypothesis [30], we predicted that wolves and dogs would form reputations of humans in a similar way. However, the Emotional Reactivity Hypothesis [13] postulates that dogs acquired advanced sociocognitive skills as a result of domestication, thus an alternative prediction is that dogs would outperform wolves.

## Materials and methods

### Ethical statement

Ethical approval was obtained from the 'Ethik und Tierschutzkommission' of the University of Veterinary Medicine Vienna (Protocol #ETK-084/05/2020). All study animals were housed at the WSC in Wildpark Ernstbrunn, Austria, and had previously participated in behavioural tests, thus they were accustomed to participating in research while separated from their pack members. The animals' participation was voluntary; if they were not motivated to leave their home enclosure or stressed during the experiment, the session was cancelled and repeated on another day. The individuals pictured in S1–S4 Video have provided written informed consent (as outlined in PLOS consent form) to publish their image alongside the manuscript, or we de-identified images by blurring faces when we were unable to get written informed consent due to unavailability.

### Subjects

At the time of testing, there were 14 adult wolves and six adult dogs living at the WSC. Five wolves were excluded (three did not participate in behavioural tests with unfamiliar humans and two did not want to participate in this study), therefore nine wolves (6 males, 3 females, $M = 8$, $SD = 2.60$) and six dogs (4 males, 2 females, $M = 7$, $SD = 1.55$) participated in the experiment (see Table 1).

The wolves were born in captivity and two dogs (Layla and Zuri) were obtained from an animal shelter in Hungary; the other dogs were born at the WSC. All animals were hand-raised by trainers from 10 days old until 5 months old and then were integrated into existing conspecific packs and lived in outdoor enclosures. Throughout their upbringing, the animals had regular but not continuous contact with the hand-raisers' pet dogs, which gave the wolves and dogs security when they were puppies and helped them to become socialised. Since the WSC animals have established close relationships with these pet dogs and submit to them, they have been used as demonstrators in previous studies [31,34,38] and were used in this study (for more details on the animals' upbringing and our reasoning for using the term 'conspecific' for dog demonstrators for both wolves and dogs, see [38]).

**Table 1. List of subjects' participation.**

| Species | Animal | Sex | Age (years) | Dog demonstrator | Condition 1 | Condition 2 | First condition |
|---------|--------|-----|-------------|------------------|-------------|-------------|-----------------|
| Wolves | Amarok | M | 8 | Freya | OTE | OTE | Control |
| | Kenai | M | 10 | Freya | OTE | OTE | Control |
| | Maikan | M | 4 | Hakima | OTE | OTE | Control |
| | Taima | F | 4 | Pepeo | Excluded | OTE | Control |
| | Chitto | M | 8 | Pepeo | OTE | OTE | Experimental |
| | Tala | F | 8 | Hakima | OTE | OTE | Experimental |
| | Geronimo | M | 11 | Freya | OTE | NTE | Experimental |
| | Yukon | F | 11 | Freya | OTE | NTE | Experimental |
| | Wamblee | M | 8 | Freya | HE | HE | Experimental |
| Dogs | Hiari | M | 6 | Koda | OTE | OTE | Control |
| | Layla | F | 9 | Rico | OTE | OTE | Control |
| | Zuri | M | 9 | Zazu | OTE | OTE | Control |
| | Imara | F | 6 | Koda | OTE | OTE | Experimental |
| | Panya | F | 6 | Rico | OTE | OTE | Experimental |
| | Enzi | M | 6 | Zazu | OTE | OTE | Experimental |

OTE = Old Test Enclosure, NTE = New Test Enclosure, HE = Home Enclosure.

The animals were fed according to their regular feeding regimes (for more details, see [39]), thus they were not food-deprived before the experiment and had ad libitum access to drinking water in their home enclosure and in the test enclosure. All animals completed the experiment between June and December 2020 except two wolves (Kenai and Taima) who completed testing in April and May 2021.

## Experimental design

There were two conditions:

1. Experimental: the subject observed two unfamiliar humans (henceforth 'partners') interact with a familiar pet dog (henceforth the 'dog demonstrator'). The trainers selected three pet dogs (Freya, Hakima and Pepeo) to act as demonstrators for the wolves and three other pet dogs (Koda, Rico and Zazu) to act as demonstrators for the dogs (see Table 1). The dog demonstrator was fixed within-subjects and each animal was paired with a pet dog with whom they had a close relationship so the subject would pay attention to the third-party interactions.

2. Control: the subject observed the partners perform the same actions as in the experimental condition without a dog demonstrator present, which was instead replaced by a red bowl. This was conducted so that if eavesdropping was observed in the experimental condition, we would be able to discern whether the animals' responses were due to the interactions between the partners and the dog demonstrator or whether the partners' actions *per se* were sufficient to allow a discrimination between them. The trainer was also absent for two reasons: first, if the trainer stood in the test enclosure, the animals might have perceived it as a third-party interaction between the humans. Second, the presence of the trainer might have distracted the subject from watching the partners, as they have a close bond with her and they know that trainers always have food in their pockets.

This was a repeated measures design, so half of the wolves and dogs experienced the experimental condition first and the other half experienced the control condition first. One wolf

(Taima) was excluded from the first condition because she showed signs of stress during the experimental procedure. There was a break of 11 days minimum between conditions.

In each condition, two unfamiliar females acted as the partners. We recruited two pairs of partners that were supposed to remain stable within conditions throughout the experiment (i.e., MP and NF in Condition 1 and CR and PB in Condition 2). However, CR and PB were unavailable after 2020, so we recruited two new partners (KC and DK) to test two wolves (Kenai and Taima) in Condition 2 in 2021. Thus, there were a total of six human partners in the experiment.

The partners wore contrasting clothes to make their distinguishing features more salient. In Condition 1, one partner wore white clothes and the other wore black clothes, and in Condition 2, one partner wore a camouflage patterned jacket and the other wore a white and pink patterned jacket. The partners' role and the colour of the clothes they wore were randomised between- and fixed within-subjects. HLJ was the main experimenter for Condition 1 and MP was the main experimenter for Condition 2. Finally, in each session, a trainer led the animals to the specific spots required by the test procedure with a dog lead.

## Experimental setup

The experiment took place in one of three locations at the WSC: the Old Test Enclosure (702 m²), New Test Enclosure (620 m²) or the home enclosure for one wolf (Wamblee, see S1 File). The subjects observed the third-party interactions from a separate compartment that was adjacent to the test enclosure (henceforth the 'observer's area'), which was 19 m² in the Old Test Enclosure and 18 m² in the New Test Enclosure.

Inside the test enclosure, the subject/dog demonstrator's starting point was 5 m from the fence, which was marked by placing a big stone on the spot. Outside the test enclosure, three crosses were marked on the ground using spray paint; the central cross was marked in parallel to the starting point and a cross was marked on either side of the central cross, 2 m away from it (hereafter referred to as P1 and P2). Two lines were dug in the ground 2 m in front of P1 and P2 inside the test enclosure to indicate when the animal had made a choice response. There was a GoPro Hero 4 Black (overview camera) placed on a tripod that recorded the whole experiment in every session (Fig 1).

## Procedure

The experiment consisted of four sessions (described below) and generally followed that of Jim et al. [40]. We conducted sessions on separate days with a break of 2 days minimum between each session, but due to unavoidable logistical problems, some animals experienced Sessions 1 and 2 on the same day and Sessions 3 and 4 on the same day with a 2-hour break between sessions and a break of 6 days between the two testing days (see Fig 2 for an overview of the full procedure). The subject could explore the test enclosure for approximately 5 minutes before every session.

**Session 1: Baseline.** This session tested whether the subject preferred one partner prior to observing any third-party interactions. The trainer entered the enclosure, stood at the starting point (5 m away from the fence), and held the subject by his/her collar. The partners walked up to the fence outside of the test enclosure and stood on P1 and P2 with their backs turned to the enclosure (their positions were randomised and counterbalanced across subjects and conditions), each holding a piece of raw meat in their hand. When the subject was ready, the trainer said, "ok" and the partners turned around (or looked up, see habituation and exceptions in S1 File), held the meat up in front of them and waved their hand for three seconds to attract the subject's attention, and then stood still and did not make eye contact with the

## (A) Old Test Enclosure

## (B) New Test Enclosure

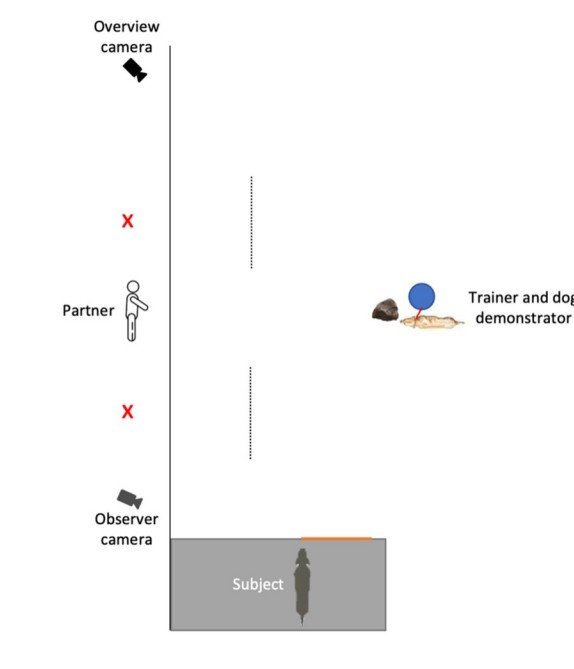

**Fig 1.** Schematic depictions of the experimental setup in the (A) Old Test Enclosure and (B) New Test Enclosure. Orange lines depict door openings. The areas shaded in grey depict the observer's area.

subject. Then, the trainer let go of the subject's collar so he/she could choose whom to approach, and the main experimenter (HLJ/MP) started the stopwatch. The trainer looked down and did not make eye contact with the animal throughout the trial so she did not influence the subject's choice, and kept her palms open to show that she had no food. If the subject walked towards a partner and his/her front paws crossed the line that was dug in the ground 2 m inside the test enclosure and his/her head was oriented towards the partner, this was considered as a choice and she threw the meat into the enclosure. After the subject ate the food, the other partner who was not chosen called the subject to get his/her attention and threw the meat into the enclosure for the subject to eat to ensure he/she did not develop a preference for one partner. If the subject did not approach either partner within one minute, it was considered a 'no choice' response; the main experimenter said, "ok" and the partners turned around, keeping the food in their hands, and the trainer called the subject back to the starting point. After this single trial, Session 1 was over; the partners left the testing area and the subject was moved back to his/her home enclosure (S1 Video).

**Session 2: Indirect experience Phase I.** Before the session began, another GoPro Hero 4 Black (observer camera) attached to a tripod was placed close to the observer's area to record whether the subject was paying attention to the third-party interactions (in the Old Test Enclosure, the observer camera was placed on P1, so a dummy tripod was placed on P2 to ensure the subject did not develop a side bias because of the presence of a tripod, see Fig 1A). After the subject explored the test enclosure for approximately 5 minutes, he/she was shifted into the

| Condition 1 | | | | | | |
|---|---|---|---|---|---|---|
| Session 1 | | Session 2 | | Session 3 | | Session 4 |
| **Baseline** | Break (min. 2 days) | **Indirect experience Phase I**<br>Observation phase<br>1. Selfish<br>2. Generous<br>3. Selfish<br>4. Generous<br>5. Selfish<br>6. Generous<br>7. Selfish<br>8. Generous<br>Test (single trial) | Break (min. 2 days) | **Indirect experience Phase II**<br>Observation phase<br>1. Generous<br>2. Selfish<br>3. Generous<br>4. Selfish<br>5. Generous<br>6. Selfish<br>7. Generous<br>8. Selfish<br>Test (single trial)<br>**Direct experience Phase I**<br>Test (6 trials) | Break (min. 2 days) | **Direct experience Phase II**<br>Experience phase<br>1. Selfish<br>2. Generous<br>3. Selfish<br>4. Generous<br>5. Selfish<br>6. Generous<br>7. Selfish<br>8. Generous<br>Test (6 trials) |

| Break (min. 11 days) | | | | | | |
|---|---|---|---|---|---|---|

| Condition 2 | | | | | | |
|---|---|---|---|---|---|---|
| Session 1 | | Session 2 | | Session 3 | | Session 4 |
| **Baseline** | Break (min. 2 days) | **Indirect experience Phase I**<br>Observation phase<br>1. Generous<br>2. Selfish<br>3. Generous<br>4. Selfish<br>5. Generous<br>6. Selfish<br>7. Generous<br>8. Selfish<br>Test (single trial) | Break (min. 2 days) | **Indirect experience Phase II**<br>Observation phase<br>1. Selfish<br>2. Generous<br>3. Selfish<br>4. Generous<br>5. Selfish<br>6. Generous<br>7. Selfish<br>8. Generous<br>Test (single trial)<br>**Direct experience Phase I**<br>Test (6 trials) | Break (min. 2 days) | **Direct experience Phase II**<br>Experience phase<br>1. Generous<br>2. Selfish<br>3. Generous<br>4. Selfish<br>5. Generous<br>6. Selfish<br>7. Generous<br>8. Selfish<br>Test (6 trials) |

**Fig 2. Flowchart illustrating an example of the whole procedure for one subject.**

observer's area. Then, the main experimenter started the cameras. This session consisted of an observation phase (described below) and a test phase, which was similar to the baseline.

In the observation phase of the experimental condition, the trainer entered the test enclosure with a dog demonstrator (familiar pet dog) on a lead and stood at the starting point. One partner walked up to the fence and stood at the central cross with a piece of meat in her hand. The trainer and the dog demonstrator walked 4 m forwards towards the partner, who then held the meat up in front of her to attract the dog demonstrator's attention. If the dog demonstrator did not look at the meat, the partner called the dog demonstrator's name. Then, the subject witnessed one of the following scenarios depending on which partner the dog demonstrator interacted with:

a. Generous: she said, "here you go!" in a friendly tone and threw the meat into the enclosure so the dog demonstrator could eat it. After the dog demonstrator had eaten it, the trainer walked the dog demonstrator back to the starting point and the partner left the testing area.

b. Selfish: she said, "you can't have it!" in an unfriendly tone, crossed her arms and turned around, keeping the food in her hand. After a few seconds, the trainer walked the dog demonstrator back to the starting point and the partner left the testing area.

The procedure was repeated with the second partner and repeated four times, i.e., eight demonstrations in total. The order of interactions was counterbalanced across subjects, i.e., the generous partner started for half of the subjects and the selfish partner started for the other half. After the observation phase, the dog demonstrator left the testing area and the main experimenter removed the tripods.

In the control condition, the only difference in the observation phase was that the trainer and the dog demonstrator were absent and a red bowl was placed 1 m from the fence in the enclosure instead. Thus, the partners 'interacted' with an invisible dog demonstrator and the subject witnessed them behave the same way as in the scenarios described above. The bowl was placed there as a target for the generous partner to throw the food into the enclosure because the trainer entered the enclosure and had to quickly find and collect the four pieces of meat that the generous partner threw into the enclosure after the demonstrations. The trainer placed them in the bowl and handed it to the main experimenter, who also removed the tripods at the end of the observation phase. Then, the subject was shifted into the test enclosure to participate in the test phase. The interval between the phases was approximately 5 minutes on average.

The test phase was similar to the baseline, but the partners' positions were counterbalanced for half of the sample (e.g., if the generous partner stood on P1 in the baseline, she stood on P2 in the test phase). If the subject walked towards a partner and his/her front paws crossed the line that was dug in the ground 2 m inside the test enclosure (Fig 1) and his/her head was oriented towards the partner, this was considered as a choice and she acted the same way as she did in the observation phase, i.e., the generous partner fed the subject and the selfish partner did not. Then, both partners turned around to indicate that they would not interact with the subject anymore. After this single trial, Session 2 was over; the partners left the testing area and the subject was moved back to his/her home enclosure (S2 Video).

**Session 3: Indirect experience Phase II & direct experience Phase I.** The subject was tested 2–8 days later and the procedure was identical to Session 2 with some minor changes. Before the session began, seven pieces of raw meat were placed into two hip bags, which were to be used in the test phase. In the observation phase, the order of the partners was counterbalanced (e.g., if the generous partner started in Session 2, then the selfish partner started in Session 3). After the observation phase, the partners each wore a hip bag. There were seven trials in the test phase, in which the partners' positions were semi-randomised so that they never stayed in the same position more than twice in a row. If the subject chose the generous partner, she fed the subject and rebaited herself by taking another piece of meat from her hip bag (S3 Video). We considered the first trial as testing for eavesdropping, as the subjects only had a brief direct experience with one partner in Session 2, and the latter six trials as testing for direct reputation formation.

**Session 4: Direct experience Phase II.** The procedure was identical to Session 3 with some minor changes. Before the session began, six pieces of raw meat were placed into two hip bags, which were to be used in the test phase. Instead of an observation phase, there was an experience phase, where the subject directly interacted with the two partners and the order of the partners was counterbalanced again (e.g., if the selfish partner started in Session 3, the generous partner started in Session 4). After the experience phase, the partners each wore a hip bag and there were six trials in the test phase instead of seven (S4 Video).

## Behavioural analysis

The subject must pay attention to the partner's actions to understand their different roles, so we coded the subject's attentiveness during the observation phase in Sessions 2 and 3 from the observer camera footage, which was synchronised and merged into one video with the overview camera footage to ease coding.

We defined the beginning of the interaction as the moment the partner raised her hand and the end as when the partner left the testing area. During the generous interaction, we coded whether the subject was attentive the moment the partner threw the food into the enclosure or

when the dog demonstrator ate the food. During the selfish interaction, we coded whether the subject was attentive the moment the partner turned around to face away from the enclosure.

We coded whether the subject was attentive or not in each interaction during each observation phase as a binary variable (yes/no). We defined the subject as being attentive when his/her head was oriented towards the direction of the interaction; any other behaviour was defined as not attentive. If the subject was out of view, we coded it as NA.

We also coded the subject's choice in the baseline and the test phase. If the subject walked towards a partner and his/her front paws crossed the line that was dug in the ground 2 m inside the test enclosure (Fig 1) and his/her head was oriented towards the partner, this was considered as a choice (generous/selfish). If the subject did not approach either partner within one minute, it was considered a 'no choice' response, which we coded as NA.

## Statistical analysis

All analyses were performed using R Statistical Software (v4.0.4; [41]) in RStudio (v2022.02.3 +492; [42]). We conducted generalised linear mixed-effect models (GLMMs) with binomial error structure and logit link function [43], which were fitted using the function glmer of the R package "lme4" (v1.1.28; [44]). For all models, we z-transformed covariates to a mean of 0 and a standard deviation of 1 and changed the optimizer used by the function glmer to "bobyqa" [45] to ease convergence [46]. We assessed model stability by comparing the estimates obtained from the model based on all data with those obtained from models with the levels of the random effects excluded one at a time. We determined the confidence of model estimates using the function confint.merMod (method = "boot") of the R package "lme4". We checked for collinearity by determining Variance Inflation Factors (VIF) [47] for a standard linear model excluding the random effects using the function vif of the R package "car" (v3.0.12; [48]). Random slopes were identified and model stability was assessed using functions kindly provided by Roger Mundry. The boxplot was created in the R package "ggplot2" (v3.3.5; [49]).

**Attentiveness.** First, we checked inter-observer reliability for attentiveness coding. HLJ, MP and a research assistant independently coded 20% of the videos, which were randomly chosen. We used Fleiss' Kappa [50] from the R package "irr" (v0.84.1, [51]) and inter-rater agreement was excellent (K = .92, $p < .001$). MP and a research assistant then each coded half of the videos.

Afterwards, we conducted a GLMM to test whether the dogs or wolves were attentive to the demonstrations in the observation phase. 472 observations were made with 15 individuals; 22 of these were NA (S1 Dataset). The response variable was whether the animal was attentive to the demonstration (yes/no). The test predictors were species (factor with two levels: dog and wolf), partner (factor with two levels: generous and selfish) and condition (factor with two levels: control and experimental). The control predictors were session (covariate) and trial (covariate). Therefore, the full model included an interaction between species × partner × condition (and all lower order terms this encompasses) and session and trial were added as fixed effects. Subject ID was added as a random intercept and trial as a random slope within subject ID. Then, we compared the full model to the null model, from which we removed species, partner, condition, and their interaction from the fixed effects part of the model. The model was relatively stable (see the range of estimates in Table 2) and there was no collinearity (max VIF = 1.002).

To test whether the dogs and wolves formed reputations of humans, we conducted three GLMMs. For all models, the response variable was the subject's choice to approach a partner (generous/selfish). Dog demonstrator ID was not included as a random effect in any of the models since the dog demonstrator was absent in half of the trials (i.e., in the control

**Table 2. Results of the full model for attentiveness.**

| Term | Estimate | SE | 95% CI | | z | p | Min | Max |
|---|---|---|---|---|---|---|---|---|
| | | | Upper | Lower | | | | |
| Intercept | 1.765 | 0.490 | 2.859 | 0.950 | | | 1.523 | 2.281 |
| Species: Wolf[a] | 0.120 | 0.644 | 1.323 | -1.211 | 0.186 | .853 | -0.428 | 0.593 |
| Partner: Selfish[b] | 0.343 | 0.601 | 1.554 | -0.645 | 0.571 | .568 | -0.025 | 0.733 |
| Condition: Experimental[c] | -1.057 | 0.525 | -0.234 | -2.199 | -2.014 | **.044** | -1.347 | -0.840 |
| z-transformed trial | 0.037 | 0.115 | 0.299 | -0.209 | 0.324 | .746 | -0.022 | 0.117 |
| z-transformed session | -0.021 | 0.112 | 0.216 | -0.238 | -0.191 | .849 | -0.059 | 0.031 |
| Species × partner | -1.539 | 0.747 | -0.491 | -2.922 | -2.061 | **.039** | -1.952 | -1.173 |
| Species × condition | -0.638 | 0.68 | 0.469 | -1.833 | -0.939 | .348 | -1.005 | -0.316 |
| Partner × condition | -0.099 | 0.776 | 1.046 | -1.353 | -0.127 | .899 | -0.785 | 0.545 |
| Species × partner × condition | 0.655 | 0.963 | 2.182 | -0.688 | 0.681 | .496 | 0.01 | 1.344 |

Estimate, standard error, confidence intervals, results of significance tests (Wald's z approximation) and the range of estimates derived after excluding individuals one at a time. Significant p values are in bold.

[a]Species: Dog as reference level.

[b]Partner: Generous as reference level.

[c]Condition: Control as reference level.

condition). We also counterbalanced and hence controlled for the partners' role, colour of clothes and positions, so we did not add them as control predictors in any of the analyses to reduce the complexity of the models.

We split the data (S2 Dataset) into three subsets and fitted three models:

**(1) Eavesdropping subset**

The eavesdropping subset comprised the baseline in Session 1 (Trial 1), the single trial in Session 2 (Trial 2) and the first trial in Session 3 (Trial 3) in the experimental and control condition to test whether dogs and wolves formed a reputation of the humans based on their indirect experience. We argue that the first trial of Session 3 is still based on observation rather than the brief direct experience in Session 2. We compared the animals' choices in the experimental and control condition to test whether their responses were due to the subject observing the social interaction between partner and dog demonstrator, or whether the partners' actions *per se* (throwing the food into the enclosure or turning away from the enclosure) were sufficient to allow a discrimination between them.

In this subset, 89 observations were made with 15 individuals; 14 of these were NA. The test predictors were trial (factor with three levels: Trials 1–3 in S2 Dataset), condition, and species. The control predictors were condition order (covariate) and attentiveness (covariate), which was determined by the proportion of interactions the animals were attentive in the observation phase of Session 2 and 3 separately. Therefore, the full model included an interaction between trial × condition × species (and all lower order terms this encompasses). Condition order and attentiveness were added as fixed effects and subject ID as a random intercept. Then, we compared the full model to the null model, from which we removed trial, condition, species, and their interaction from the fixed effects part of the model. The model was very unstable (see the range of estimates in S1 Table) and attentiveness was slightly collinear (VIF = 2.167) because there was no observation phase in the baseline (Trial 1), thus attentiveness was 0 for all subjects.

**(2) Reputation-learning subset**

The reputation-learning subset comprised the latter six trials of Session 3 (Trials 4–9 in S2 Dataset) to test whether dogs and wolves formed a reputation of the humans based on their very limited indirect and direct experience. In this subset, 174 observations were made with 15 individuals; 14 of these were NA. The test predictors were condition, species, and trial, which was z-transformed (covariate). The full model included an interaction between condition × species × trial (and all lower order terms this encompasses). Condition order and attentiveness were added as fixed effects. Subject ID was added as a random intercept and trial as a random slope within subject ID. Then, we compared the full model to the null model, which lacked condition, species, trial, and the interaction in which they were involved. Model stability was good (see the range of estimates in S2 Table) and there was no collinearity (max VIF = 1.614).

### (3) Direct experience subset

The direct experience subset comprised the six trials of Session 4 (Trials 10–15 in S2 Dataset) to test whether dogs and wolves formed a reputation of the humans based on their more extensive direct experience. In this subset, 174 observations were made with 15 individuals; 11 of these were NA. The test predictors were species and trial, which was z-transformed (covariate). The full model included an interaction between species × trial (and all lower order terms this encompasses) and condition order was added as a control predictor. Condition was not included as a test predictor and attentiveness was not included as a control predictor because these were only relevant for testing eavesdropping. Subject ID was added as a random intercept and trial as a random slope within subject ID. Then, we compared the full model to the null model, where species and trial and their interaction was removed from the fixed effects part of the model. The model was very stable (see the range of estimates in S3 Table) and there was no collinearity (max VIF = 1.002).

## Results

### Attentiveness

The likelihood ratio test comparing the full and null model revealed that the test predictors (species, partner and condition) had a significant effect on the subjects' attentiveness in the observation phase ($\chi^2 = 48.270$, $df = 7$, $p < .001$). Specifically, dogs and wolves were significantly more attentive towards both partners in the control condition than the experimental condition ($p = .044$). Further, the interaction between species × partner shows that dogs were equally attentive to the generous and selfish partner but wolves were significantly more attentive to the generous partner than the selfish one in both conditions ($p = .039$) (Fig 3, Table 2).

### (1) Eavesdropping subset

The likelihood ratio test comparing the full and null model revealed that the interaction between trial × condition × species did not have a significant effect on the subjects' choice for the generous partner over the selfish partner ($\chi^2 = 15.972$, $df = 11$, $p = .142$, S1 Table).

### (2) Reputation-learning subset

The likelihood ratio test comparing the full and null model revealed that the interaction between condition × species × trial did not have a significant effect on the subjects' choice for the generous partner over the selfish partner ($\chi^2 = 4.376$, $df = 7$, $p = .736$, S2 Table).

### (3) Direct experience subset

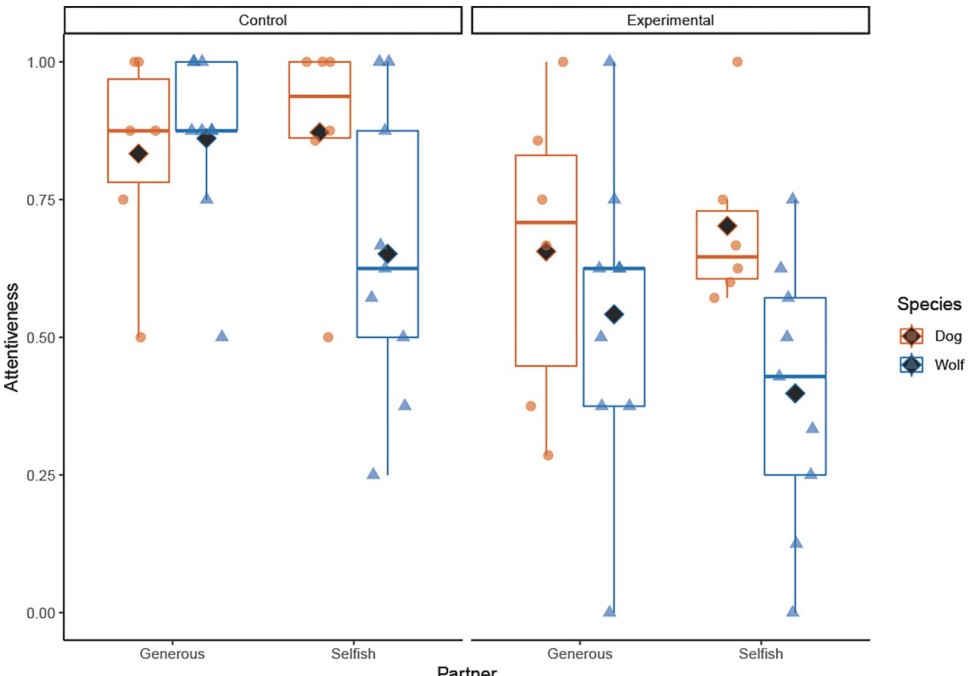

**Fig 3. Dogs' and wolves' attentiveness towards the generous and selfish partner in the observation phase in the control and experimental condition.** The black diamond depicts the mean.

The likelihood ratio test comparing the full and null model revealed that the interaction between species × trial did not have a significant effect on the subjects' choice for the generous partner over the selfish partner ($\chi^2 = 0.687$, $df = 3$, $p = .876$, S3 Table).

## Exact binomial tests

As the results of the GLMMs were non-significant, we conducted exact binomial tests [52] to test whether each subject showed a side or partner bias in Conditions 1 and 2 grouped together. We counted the number of left choices as success for side. The partners' role and colour of clothes were randomised between- and fixed within-subjects, so we counted the number of times the subject chose the generous partner as success. 0.5 was the hypothesised probability of success for both.

The results showed that 4 out of 9 wolves and 2 out of 6 dogs showed a significant side bias for the right (S4 Table). Additionally, two wolves (Tala and Yukon) and three dogs (Enzi, Imara and Layla) significantly preferred the generous partner (Fig 4, S4 Table). Interestingly, the generous partner that the animals showed a preference for were the same two individuals (NF in Condition 1 and CR in Condition 2).

## Discussion

The aim of this study was to test whether dogs and wolves can form reputations of humans after observing them interact with a dog and/or after directly interacting with them in a food-giving situation. Our main finding is that dogs and wolves, at the group level, did not differentiate between a generous and selfish partner after indirect or direct experience with the humans. Therefore, our results do not support the hypothesis that dogs and wolves can form reputations. However, the predictor 'species' was close to significance in the reputation-

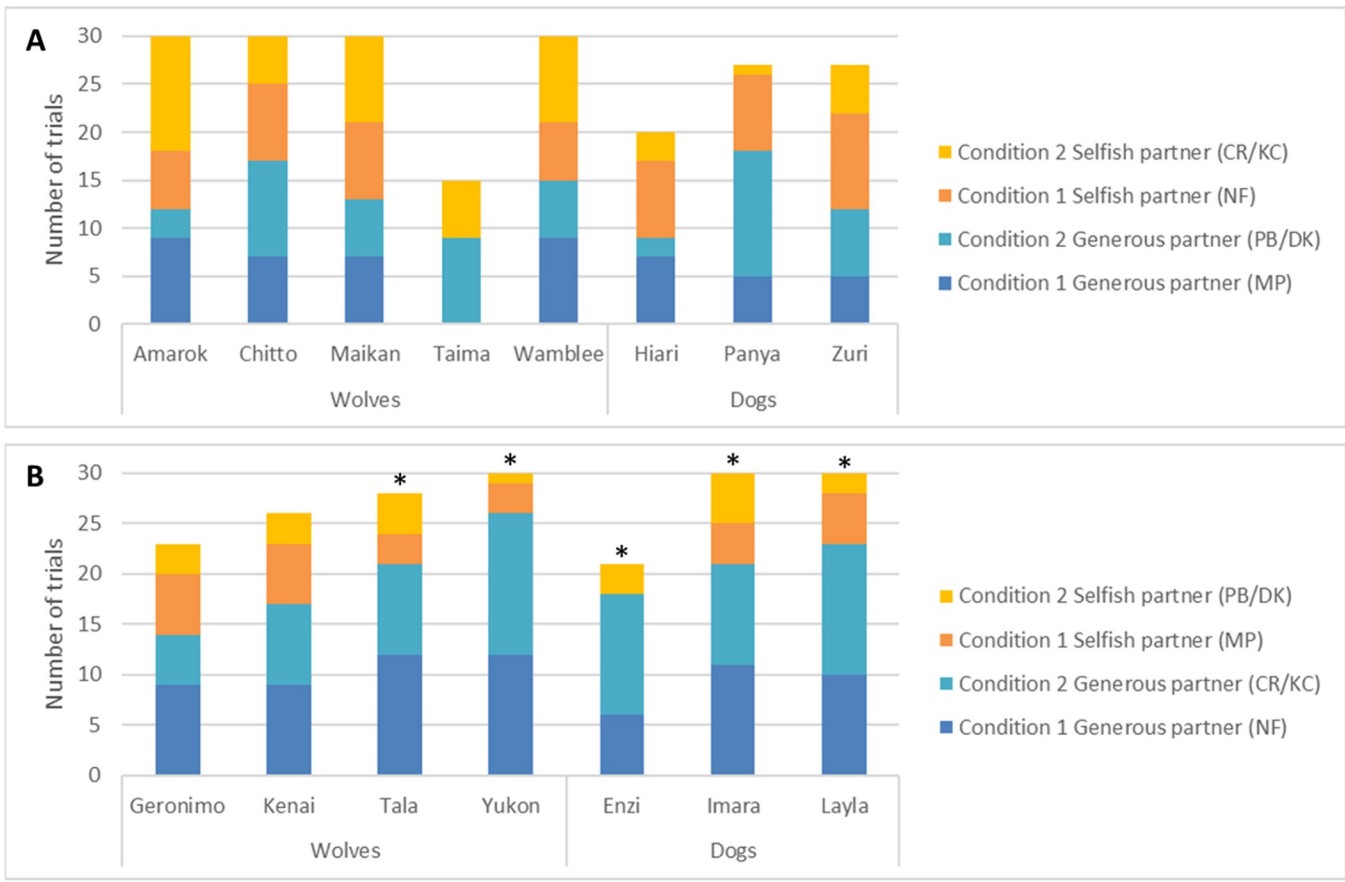

**Fig 4. Number of trials subjects chose the generous or selfish partner.** (A) shows MP/PB/DK as the generous partner and NF/CR/KC as the selfish partner for half of the sample and (B) shows the opposite for the other half of the sample. * represent significant *p* values.

learning model (*p* = .067) and the model stability was good (see the range of estimates in S2 Table), so it is possible that wolves may have preferred the generous partner after direct experience. In support of this, wolves and dogs differed regarding their attentiveness towards the two partners, with wolves being more discriminative and paying more attention to the generous partner than the selfish one in the observation phase. Moreover, five animals (two wolves and three dogs) did preferentially choose the generous partner, but this reputation was formed over all trials, so it is possible that the combined experience of observing and directly obtaining the food from the generous partner led to this discrimination.

The fact that we found no support for wolves and dogs forming indirect reputations at the group level could be due to several reasons. One is that the animals may have needed to watch more demonstrations in the observation phase to discriminate between the partners and/or form reputations of humans. Previous studies have found that dogs can form reputations after observing 3–5 interactions with each partner [21–24]. In our study, the animals were exposed to four interactions per partner in a single session, which is within the range of the previous studies, and we did not increase the number of the demonstrations for two reasons. First, the wolves would have been stressed if they were kept in the observer's area for longer, and second, it is likely that they would have lost interest and not paid attention to the demonstrations anymore. Nevertheless, it is possible that we may have found evidence for indirect and direct reputation formation in wolves and dogs if they had observed more demonstrations/experiences in

the observation/experience phase, respectively, and more trials in the test phase. Therefore, we replicated this study with pet dogs and increased the number of demonstrations/experiences from four to six per partner in the observation/experience phase and the number of trials from six to twelve in the test phase, but we still found no evidence of indirect or direct reputation formation (unpublished). Having said that, we did find that five animals (two wolves and three dogs) in this study significantly preferred the generous partner over the selfish one when analysing the individual data. It is possible that these individuals inherently preferred a certain person, since the same two humans (NF and CR) acted as the generous partner for all of them, but we think this is unlikely for two reasons. First, the animals experienced two different humans act as the generous partner in the two conditions (i.e., NF in Condition 1 and CR in Condition 2). Second, none of them showed a bias for NF or CR in the baseline. Therefore, we instead suggest that some animals were faster at discriminating between the two partners' actions and they chose the generous partner because of her behaviour rather than her identity, and others needed more demonstrations/experiences to form reputations. This is further supported by Subiaul et al. [8]–chimpanzees largely varied in the number of direct experiences they needed before maintaining a significant preference for the generous partner (32–184 trials).

Another possible explanation why the animals did not form reputations of the humans is that they could not distinguish between the generous and the selfish partners' actions. This could account for the negative results in Jim et al. [27], where both partners performed the same pushing action and only the outcome was different, but in this study, the differences between the partners' actions were greater–the selfish partner crossed her arms and turned her back from the dog demonstrator, whereas the generous partner did not turn her back from the demonstrator and threw the piece of meat into the enclosure. The dog demonstrator's reactions to the partners were different too–the dog ate the food after interacting with the generous partner and did not eat after interacting with the selfish partner.

The differences in the partners' actions are important for the animals to be able to discriminate between them, but the fact that the selfish partner kept the food in her hand may have been a potential confound, as the longer duration of the presence of the food with the selfish partner may have led some animals to choose her. Therefore, the selfish partner's actions may have competed with the presence of food and overshadowed a clear preference for one partner, which may explain our non-significant results. It would have been better to have the selfish partner act 'competitively' and eat the food in front of the animal like in Heberlein et al.'s studies [16,37], but this was not possible because we used pieces of raw meat as rewards to increase the animals' motivation, thus the selfish partner could not eat it.

Our sample size was very small, which limited the power of our statistical analyses. Studies investigating reputation formation in animals often have small sample sizes (e.g., there are between 4 and 17 individuals in [5,8–10,40,53]) and it can be difficult to interpret the results because conducting a GLMM may be problematic if there are not enough data to measure the response accurately, or there may not be enough power to analyse the results adequately and thus show false positive or false negative results. To counteract this problem, we also conducted exact binomial tests to analyse the behaviour of each subject, which indeed showed that some individuals behaved in a way consistent with forming a reputation of the two partners.

Aside from the non-significant results of eavesdropping, we found that dogs and wolves were more attentive in the control condition (where a red bowl was placed inside the enclosure instead of the dog demonstrator) compared to the experimental condition, which may indicate that the latter condition was not interesting enough. The animals may have been more attentive in the control condition because the subject observed the food being thrown into the enclosure while no one was inside, thus the food remained inside the enclosure, whereas in the

experimental condition, the animals observed the dog demonstrator eat the food, thus the food was no longer available to them. Consequently, the animals may have lost interest in the third-party interactions, which may explain why we did not find evidence for eavesdropping at the group level. Another possibility is that the animals may have associated the red bowl with food because they are used regularly for feeding at the WSC. We placed the bowl in the enclosure as a target for the generous partner when she threw the meat, but it would have been better if we had used a different bowl that the animals had never seen before. Nevertheless, we found no significant difference in the animals' choice between the generous or selfish partner in the control and experimental condition, so the presence of the red bowl did not seem to have a strong influence.

Another finding of our study is that wolves were more attentive to the generous than the selfish person in the observation phase than dogs. One could speculate that dogs should pay more attention to humans than wolves because they rely on humans for valuable resources, such as food and shelter [25], but studies have shown that wolves pay more attention when observing others' behaviours or to details compared to dogs. For example, Range and Virányi [31] found that wolves can follow human and conspecific gaze and can even follow a human gaze cue into distant space, whereas dogs did not [32]. Moreover, dogs and wolves paid similar attention to a human demonstration but wolves were more attentive towards behavioural details of conspecifics [34] and more sensitive towards the details of the action demonstrated by a conspecific than dogs [38]. It has been hypothesised that being attentive to others' behaviours is more important for wolves than dogs because of their different feeding ecologies and social organisations; wolves rely on pack members for group hunting and pup-rearing, whereas dogs mostly forage alone on human refuse and show little allomaternal care (Social Ecology Hypothesis [54]). As the wolves at the WSC are hand-raised by humans, it is possible that they are equally attentive to humans and conspecifics, so the wolves may have stopped paying attention to the selfish partner if they understood that she would not provide food.

The animals at the WSC are used to seeing familiar humans (the trainers) interact with conspecifics in situations involving food, such as during shifting or training, and they are used to seeing unfamiliar humans interact with conspecifics in contexts without food during weekly pack visits. Unfamiliar humans are not allowed to have food around the WSC animals, thus we thought they would be more attentive to the unusual situation that the unfamiliar partners had pieces of meat that they could feed to them. That being said, our experimental setup still may not have been interesting enough for the animals but it is crucial for the subject to pay attention to the partners' actions to understand their different roles to form reputations of them. We coded the animals' attentiveness as a binary variable (i.e., whether the subject's head was oriented towards the direction of the interaction) since we could not code it as a continuous variable (i.e., duration) for two reasons. First, we could not place the camera closer to the subject by placing it inside the test enclosure because it may have obstructed the subject's view of the third-party interactions. Second, the observer's area was quite spacious (e.g., 19 m$^2$) to allow the wolves to move around freely, as tight spaces are stressful for them, but this meant that the animal was sometimes out of view or too far away to see whether he/she was paying attention to the interaction.

We found that some animals showed a side bias and all those who did preferred the right side, which suggests that it was due to the environment at the Old Test Enclosure. The partner standing on the left (P1) was close to the corner of the enclosure, which the animals may not have felt comfortable to approach, and the partner standing on the right (P2) was close to the entrance of the test enclosure and the animals may have preferred to approach that side, as that is where the trainers entered from. Furthermore, the water bucket where the animals could drink from was by the entrance of the test enclosure (Fig 1A), and although it did not

count as a choice to approach the partner on P2 when the animals went to the bucket to drink, it may have increased the probability of the animals to choose that partner.

Our study adds to the literature reporting no evidence of eavesdropping in dogs [15,25–27] and other animals, including Asian elephants (*Elephas maximus*) [40], cats (*Felis silvestris catus*) [53,55] and some results on non-human primates [5,8,11]. The mixed results on eavesdropping in animals may be due to the different methodologies used, for example, whether the setup was a food-giving situation (e.g., [5,8,21,22,26] or a helping situation (e.g., [23,24]) and whether the subject observed human-human (e.g., [11,25,27,55]) or human-animal interactions (e.g., [15,28,40,53]). We argue that using human-animal interactions enhanced the relevance of the interactions, especially since the WSC animals live in packs and regularly see conspecifics interacting with different people, like trainers (hand-raisers/very familiar), researchers (familiar) and visitors (unfamiliar).

While we did not find evidence of eavesdropping in this study, we cannot exclude the possibility that some animals may have in fact formed reputations of the humans but did not choose the generous person in the choice test. Previous studies on reputation formation in dogs have used other ways to measure partner preference in the test phase, such as duration of interacting and/or being in close proximity to the experimenter (e.g., [15,22,26,27]) and looking at the experimenter (e.g., [19,27]), but we could not use these measures in our study for two reasons. First, the unfamiliar partners were not allowed to be in the test enclosure with the wolves for safety reasons, so we could not measure the duration of an animal in close proximity to the partners. Second, when measuring duration, the experimenters should not interact with the subject and therefore would not feed the animal for a certain time period (e.g., one minute), which would have caused frustration in the wolves, as they are highly food-motivated. However, it is noteworthy that Jim et al. [27] found a significant difference in the first experimenter dogs looked at but not in their first approach, thus future studies may yield different results if alternative measures are used to analyse reputation formation. In Jim et al. [27], it was possible to analyse dogs' looking behaviour because the partners wore body cameras; using this measure may be particularly fruitful when studying wolves because they are more neophobic than dogs [56,57] and appear to have an ingrained fear of humans [58,59]. Therefore, in this study, the wolves may have formed reputations of the humans but were not comfortable approaching the unfamiliar partners and did not cross the line dug in the ground inside the test enclosure to indicate that they had made a choice in the test phase.

Although our finding that neither dogs nor wolves formed reputations of humans after direct experience is somewhat surprising given there is some evidence that wolves [37] and dogs [15–18,37] can form direct reputations, other studies have found that they could not [19,20]. Additionally, Jim et al. [39] could not demonstrate that Asian elephants could form reputations of humans after direct experience and Subiaul et al. [8] found that chimpanzees only learnt to discriminate between two humans after at least 32 trials, and one chimpanzee failed to show a consistent preference for a familiar generous partner. These findings suggest that direct reputation formation may be more difficult for animals than expected and questions whether animals can form reputations or whether they learn to over many repetitions. To test this, future studies could increase the number of experiences across more sessions. Even so, this is not what we would expect if reputation formation were important for survival, as direct interactions with a dangerous individual may be costly in the wild, so eavesdropping may be crucial for an animal's survival. In Marzluff et al.'s [60] study, human experimenters wore masks while catching and ringing wild American crows (*Corvus brachyrhynchos*), which was an aversive experience for the crows. They found that directly handled crows remembered the masks worn during catching and responded with significantly higher scolding intensity than towards the control masks, and nearby observer crows who were not handled did so as

well [61]. Additionally, in Blum et al. [62], a masked human experimenter walked past raven (*Corvus corax*) aviaries while holding a dead raven, and another human experimenter wearing a different mask walked past empty-handed. Then, the masked experimenters stood in front of the aviaries without the dead raven. They found that the ravens quickly learnt to distinguish between the two humans and scolded more towards the human wearing the 'dangerous' mask than the control mask. Thus, it appears that the ravens formed different reputations of the humans and they continued to scold the 'dangerous' human more than the neutral human over a 4-year period without further experimental reinforcement. Similar findings have also been shown in African elephants (*Loxodonta africana*) in a more naturalistic setting; Bates et al. [63] found that wild African elephants in the Amboseli National Park, Kenya, demonstrated more fearful behaviours towards Maasai men, who spear elephants to demonstrate their virility, compared to Kamba men, who do not. They also found that elephants with no experience of spearing showed similar reactions to those that had interacted with Maasai men before. These studies suggest that animals may be more likely to form reputations of humans in potentially dangerous situations, as eavesdropping is energetically costly. In this study, there was only a small cost of not receiving a food reward if the animals chose the selfish partner. Further, the trainer fed the subject every time he/she went back to the starting point for the next trial and the animals were not food-deprived before the test, thus they may not have been very motivated to choose the generous partner. Therefore, context may be crucial for animals to form reputations, which is difficult to test experimentally, as we cannot put animals in stressful or dangerous situations for ethical reasons.

In conclusion, our study does not provide support for dogs and wolves being capable of indirect or direct reputation formation of humans at the group level, though our data indicate that 1) wolves were more attentive towards the generous person during the observation phase and 2) some individuals did prefer the generous partner, at least after indirect and direct experience was combined. Thus, individual differences in this ability could account for our negative results at the group level and a larger sample size might have provided more conclusive results. This study was the first to test reputation formation in wolves and we compared this sociocognitive ability in equally raised and kept wolves and dogs and found that both species performed similarly. We emphasise the importance of context when studying reputation formation in animals and future studies should increase the relevance of the situation for the animals to help us answer this question.

## Supporting information

**S1 File. Habituation and exceptions.**
(DOCX)

**S1 Table. Results of the full model for the eavesdropping subset.** Estimates, standard error, confidence intervals, results of significance tests and minimum and maximum of model estimates derived after excluding individuals one at a time.
(DOCX)

**S2 Table. Results of the full model for the reputation-learning subset.** Estimates, standard error, confidence intervals, results of significance tests and minimum and maximum of model estimates derived after excluding individuals one at a time.
(DOCX)

**S3 Table. Results of the full model for the direct experience subset.** Estimates, standard error, confidence intervals, results of significance tests and minimum and maximum of model

estimates derived after excluding individuals one at a time.
(DOCX)

**S4 Table. Results of the exact binomial tests for side and partner bias.** Significant *p* values are in bold.
(DOCX)

**S1 Video. Example of the procedure of the baseline (Session 1) in the Old Test Enclosure.**
(MP4)

**S2 Video. Example of the procedure of the indirect experience Phase I (Session 2) in the experimental condition in the Old Test Enclosure.** The video shows the selfish partner (wearing white) and the generous partner (wearing black) interacting with the dog demonstrator in the observation phase, and the test phase. In the observation phase, the top left window shows footage from the overview camera and the top right window shows footage of the subject in the observer's area from the observer camera.
(MOV)

**S3 Video. Example of the procedure of the indirect experience Phase II & direct experience Phase I (Session 3) in the control condition in the New Test Enclosure.** The video shows the generous partner (wearing a camouflage patterned jacket) and the selfish partner (wearing a white and pink patterned jacket) perform the same actions as in the experimental condition but without a dog demonstrator present (and was replaced by a red bowl) in the observation phase, and the first out of seven trials in the test phase. In the observation phase, the top left window shows footage from the overview camera and the top right window shows footage of the subject in the observer's area from the observer camera.
(MP4)

**S4 Video. Example of the procedure of the direct experience Phase II (Session 4).** The video shows the subject interacting with the selfish partner (wearing black) and the generous partner (wearing white) in the experience phase and the first out of six trials in the test phase.
(MP4)

**S1 Dataset. This dataset was used to run the GLMM to test whether the dogs or wolves were attentive to the demonstrations in the observation phases.**
(CSV)

**S2 Dataset. This dataset shows the subjects' choices and was used to run the GLMMs to test whether the dogs or wolves formed reputations based on indirect and direct experience.**
(CSV)

## Acknowledgments

The Wolf Science Center (WSC), now part of the University of Veterinary Medicine Vienna, was established by Zsófia Virányi, Kurt Kotrschal and Friederike Range. We thank Nora Foerst (NF), ""Claudia Roethy (CR), Petra Broz (PB), Katharine Creagh (KC) and Denise Kubala (DK) for acting as partners in the study. We thank the WSC trainers and their pet dogs. Finally, we thank Roger Mundry and Mayte Martínez for statistical advice, Angelina Rosenstock for inter-observer reliability coding and Hannah Wadham for help with the videos in the Supporting information.

## Author Contributions

**Conceptualization:** Hoi-Lam Jim, Marina Plohovich, Sarah Marshall-Pescini, Friederike Range.

**Data curation:** Hoi-Lam Jim, Marina Plohovich.

**Formal analysis:** Hoi-Lam Jim.

**Funding acquisition:** Friederike Range.

**Investigation:** Hoi-Lam Jim, Marina Plohovich, Sarah Marshall-Pescini, Friederike Range.

**Methodology:** Hoi-Lam Jim, Sarah Marshall-Pescini, Friederike Range.

**Project administration:** Friederike Range.

**Resources:** Friederike Range.

**Supervision:** Sarah Marshall-Pescini, Friederike Range.

**Writing – original draft:** Hoi-Lam Jim.

**Writing – review & editing:** Hoi-Lam Jim, Marina Plohovich, Sarah Marshall-Pescini, Friederike Range.

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
