## [Decision Letter · Decision Letter 0]

8 Mar 2022

PONE-D-21-40254Wolves and dogs fail to form reputations of humans after indirect and direct experience in a food-giving situationPLOS ONE

Dear Dr. Jim,

Thank you for submitting your manuscript to PLOS ONE. After careful consideration, we feel that it has merit but does not fully meet PLOS ONE’s publication criteria as it currently stands. Therefore, we invite you to submit a revised version of the manuscript that addresses the points raised during the review process. We could only get one reviewer who recommended major changes. If you can make these changes, we can consider the manuscript for publication.

We look forward to receiving your revised manuscript.

Kind regards,

Mathew S. Crowther, Ph.D

Academic Editor

PLOS ONE

Journal Requirements:

Additional Editor Comments:

Almost impossible to get people to review so if you can address this reviewer's comments we can consider this manuscript for publication

Reviewers' comments:

Reviewer's Responses to Questions

**Comments to the Author**

1. Is the manuscript technically sound, and do the data support the conclusions?

Reviewer #1: Yes

2. Has the statistical analysis been performed appropriately and rigorously? 

Reviewer #1: Yes

3. Have the authors made all data underlying the findings in their manuscript fully available?

Reviewer #1: Yes

4. Is the manuscript presented in an intelligible fashion and written in standard English?

Reviewer #1: Yes

5. Review Comments to the Author

Reviewer #1: The manuscript “Wolves and dogs fail to form reputations of humans after indirect and direct experience in a food-giving situation” is focused on the ability of dogs and wolves to form reputation of unknown humans based on indirect and direct experience with them. The results obtained in previous studies assessing this skill in dogs have been controversial and wolves have not been evaluated in these tasks yet. Therefore, the topic is relevant and original. In general, the manuscript is clear and the methodology as well as the discussion are appropriate. It is also remarkable that dogs and wolves were evaluated in the same experimental setting that represents a great advantage.

However, I have some concerns that I will mention below.

In my opinion, one important factor is the low level of getting attention cues the animals received during the entire task. The authors did not report the duration of the attentive response of the animals during the observation phase, so it is hard to know which level of attention they showed. Considering that this is a complex task, the time animals spent looking at the demonstrations is crucial.

Moreover, the differences between the generous and the selfish person were subtle, with few ostensive cues, making it very difficult to distinguish between them.

L 123-124 I think that the statement “wolves and dogs have similar sociocognitive skills” is too broad, especially considering that you give examples only about cooperation.

L 136 Which is the evidence supporting this idea: “we enhanced the relevance of the third-party interactions by including a human-to-animal interaction in a food-giving situation”?. It is highly possible that there is a difference according to the species about this. For instance; for dogs it may be more relevant the human-human interaction than the human-dog interaction. In addition, it is possible that for dogs, the interaction between the human and the other dog evokes other emotions like jealousy. You must include this in the discussion.

Another important issue is that you used for both species a dog demonstrator, assuming that this difference in the demonstrator’s species has no effects on the performance of the animals. It would be necessary to justify better this equivalence.

L 141 Why did you mention only this prediction?

How frequently did you have to interrupt the evaluation due to the animals’ lack of motivation? Was it similar for dogs and wolves?

Why were animals not food-deprived? This strongly influences the effort of solving a task and the final performance of the animals.

How did you choose the dog demonstrator for each animal? All of them were equally familiar with the subjects?

How many trials were included in the baseline?

In contrast with the generous person, the selfish one kept the food in her hands. The longer duration of the presence of the food associated with the selfish human could increase the attentiveness and the preference towards her. So, the human attitude could compete with the presence of the food, overshadowing a clear preference for one person. You must include this limitation in the discussion.

Which was the interval between the observation and the test phases?

Why was the trainer absent during the control condition? She could have acted as a secure base during the experimental one, increasing the attention towards the demonstration.

Please, include the definition of the variable “subject’s choice to approach one person” in the behavioural analysis section.

“Fig 3. Dogs’ and wolves’ attentiveness in the control and experimental condition” Please add if the behavior was towards both partners and during the observational phase.

I think that the hypothesis regarding the importance of more experience (more demonstration trials) cannot be discarded. On the one hand, many works have shown that the performance of the animals improves with additional training trials, especially in difficult tasks like this. On the other hand, the fact that some animals showed a preference for the generous person could mean that they were faster discriminating both human attitudes. Therefore, the ability can be present in these species but some animals need more experience to achieve their goal.

I wonder at which level the bowls were familiar with dogs and with wolves. Was there any previous association between bowls and food?

It is fascinating that wolves are more attentive than dogs in several tasks. This difference could be related to different feeding ecologies. However, the explanation is not clear to me. Even when dogs do not depend on conspecifics to obtain food, they strongly depend on humans to do that. So, it would be expected for them to be very attentive and looked at humans during the demonstrations longer than wolves. In this experiment it is more important to pay attention to the human than to the dog demonstrator.

6. PLOS authors have the option to publish the peer review history of their article (what does this mean?). If published, this will include your full peer review and any attached files.

Reviewer #1: No

---

## [Author Response · Author response to Decision Letter 0]

14 Jun 2022

Response to Reviewer #1:

Thank you for your comments. We appreciate the time and detail you provided to improve our manuscript. I have reordered the comments so that they are in chronological order to correspond with line numbers in the manuscript.

The manuscript “Wolves and dogs fail to form reputations of humans after indirect and direct experience in a food-giving situation” is focused on the ability of dogs and wolves to form reputation of unknown humans based on indirect and direct experience with them. The results obtained in previous studies assessing this skill in dogs have been controversial and wolves have not been evaluated in these tasks yet. Therefore, the topic is relevant and original. In general, the manuscript is clear and the methodology as well as the discussion are appropriate. It is also remarkable that dogs and wolves were evaluated in the same experimental setting that represents a great advantage.

However, I have some concerns that I will mention below.

L 123-124 I think that the statement “wolves and dogs have similar sociocognitive skills” is too broad, especially considering that you give examples only about cooperation.

We have added other examples of similar sociocognitive skills in dogs and wolves but we deliberately emphasize how dogs and wolves are similar in their ability to cooperate, as eavesdropping may facilitate cooperation (line #134):

previous studies conducted at the WSC have shown that similarly raised and kept wolves and dogs with comparable experience of humans have similar sociocognitive skills – wolves are as skilled as dogs, if not better, at following human gazing cues into distant space and around barriers [31, 32] and at following human communicative cues (i.e., looking or pointing) [33], and they do not differ in their capacity to learn from human partners [34]. Even more importantly, wolves and dogs can cooperate with humans [35] and can even recruit human partners in a cooperative string-pulling task [36]. These findings provide support for the Canine Cooperation Hypothesis [30] and it is plausible that wolves and dogs may be able to eavesdrop, as it could facilitate cooperation.

L 136 Which is the evidence supporting this idea: “we enhanced the relevance of the third-party interactions by including a human-to-animal interaction in a food-giving situation”?. It is highly possible that there is a difference according to the species about this. For instance; for dogs it may be more relevant the human-human interaction than the human-dog interaction. 

We have removed the statement about enhancing the relevance of the third-party interactions in the introduction (line #155):

The aim of the current study was to test whether dogs and wolves can form direct and/or indirect reputations of humans using human-to-animal interactions in a food-giving situation.

Instead, we provided our rationale for why we think using human-animal interactions is more relevant for the animals than human-human interactions in the discussion (line #674):

We argue that using human-to-animal interactions enhanced the relevance of the interactions, especially since the WSC animals live in packs and regularly see conspecifics interacting with different people like trainers (hand-raisers/very familiar), researchers (familiar) and visitors (unfamiliar).

L 141 Why did you mention only this prediction?

Thank you for pointing this out. We have clarified our hypotheses and predictions (line #159):

Based on previous studies conducted at the WSC that showed similarly raised and kept wolves and dogs have similar sociocognitive skills with humans, we predicted that wolves and dogs would form reputations of humans in a similar way. Alternatively, the Emotional Reactivity Hypothesis [13] postulates that dogs acquired advanced sociocognitive skills as a result of domestication, thus it would be predicted that dogs outperform wolves on this skill.

Another important issue is that you used for both species a dog demonstrator, assuming that this difference in the demonstrator’s species has no effects on the performance of the animals. It would be necessary to justify better this equivalence.

Our rationale for why we used a dog demonstrator as a ‘conspecific’ demonstrator for both species is described in detail in another study by Range & Virányi (2014) that also used pet dogs as a demonstrator:

“Albeit the demonstrators for both dogs and wolves were trained pet dogs, we use the term conspecific in case of both groups in order to contrast our study with former studies that investigated the effects of domestication on the interactions of dogs and wolves with humans. From a systematic perspective, wolves and dogs are often categorized either as 2 species, 2 subspecies or 2 forms - domesticated and wild - of the same species. Either choice would probably provide an oversimplified static picture; dogs and wolves are rather at the beginning of the evolutionary process of a potential speciation. It is reflected in the fact that although their living environment and to some extent their behavior differ, dogs and wolves are able to form mixed groups in which they establish social relationships [31], but they can also interact with each other even without being socialized with members of the other ‘‘species’’ (e.g. can mate and produce offspring together) [32,33].”

Therefore, we have added a sentence that refers to Range & Virányi (2014) to avoid repetition (line #186):

Throughout their upbringing, the animals had regular but not continuous contact with the hand-raisers’ pet dogs, which gave the wolves and dogs security when they were puppies and helped them to become socialized. Since the WSC animals have established close relationships with these pet dogs and submit to them, they have been used as demonstrators in previous studies [31, 34, 38] and were used in the current study (for more details on their upbringing and our reasoning for using the term ‘conspecific’ for dog demonstrators for both wolves and dogs, see [38]).

Why were animals not food-deprived? This strongly influences the effort of solving a task and the final performance of the animals.

We did not deprive the animals of food for ethical reasons and we have added a sentence that refers to Rao et al. (2018), which describes the WSC animals’ regular feeding regimes in detail (line #193):

The animals were fed according to their regular feeding regimes (for more details, see [39]) and were not food-deprived before the experiment

None of the previous studies on eavesdropping or reputation formation in dogs that required them to complete choice tests involved food deprivation either – one possible reason for that is because if they were food-deprived, they may have been too fixated on the food and not paid attention to the behaviour of the partners. This was observed in Nitzschner et al.’s (2012) pilot study that investigated reputation formation in pet dogs (which were not food-deprived): “In this pilot phase, we found that the dogs did not develop a preference for the ‘giving donor’, even after many direct experiences. A possible explanation for this could be that the dogs focused their attention on the food more than on the behavior of the experimenters.” Further, there was no task to be solved in our study so we do not think it influenced the animals’ choices.

How did you choose the dog demonstrator for each animal? All of them were equally familiar with the subjects?

We have added this information in line #205:

The trainers selected three familiar pet dogs (Pepeo, Hakima and Freya) to act as demonstrators for the wolves and three other familiar pet dogs (Zazu, Koda and Rico) to act as demonstrators for the dogs (see Table 1). The dog demonstrator was fixed within-subjects and each animal was paired with a pet dog with whom they had a close relationship so the subject would pay attention to the third-party interactions.

Why was the trainer absent during the control condition? She could have acted as a secure base during the experimental one, increasing the attention towards the demonstration.

We have added our rationale for the absence of the trainer in the control condition (line #217):

The trainer was also absent for two reasons: first, if the trainer stood in the test enclosure, the animals might have perceived it as a third-party interaction between the humans. Second, the presence of the trainer might have distracted the subject from watching the partners, as they have a close bond with her and they know that the trainer had food in her pockets.

How many trials were included in the baseline?

Only one, which we have stated clearly now in line #290:

After this single trial, Session 1 was over and the partners and the subject left the testing area (S1 Video).

Which was the interval between the observation and the test phases?

We have clarified this in line #320:

After the observation phase, the dog demonstrator left the testing area, the main experimenter removed the tripods, and the subject was shifted into the test enclosure to participate in the test phase. The interval between the phases was approximately 5 minutes on average.

I wonder at which level the bowls were familiar with dogs and with wolves. Was there any previous association between bowls and food?

Thank you for this comment. First, we have explained why the bowl was used in the control condition (line #336):

The bowl was placed there as a target for the generous partner to throw the food into the enclosure because the trainer entered the enclosure and had to quickly find and collect the four pieces of meat that the generous partner threw into the enclosure. The trainer placed them in the bowl and handed it to the main experimenter, who then also removed the tripods at the end of the observation phase.

And in response to your comment: the animals did know the red bowls prior to our study but we do not think it had a strong influence on our results. We have addressed this point in the discussion (line #619):

Another possibility is that the animals may have associated the red bowl with food because they are used regularly for feeding at the WSC. We placed the bowl in the enclosure as a target for the generous partner when she threw the meat, but it would have been better if we had used a different bowl the animals had never seen before. Nevertheless, we found no significant difference in the animals’ choice between the generous or selfish partner in the control and experimental condition, thus the presence of the red bowl did not seem to have a strong influence.

Please, include the definition of the variable “subject’s choice to approach one person” in the behavioural analysis section.

We have now included the definition of the subject’s choice to approach a partner in the behavioural analysis section (line #382):

We also coded the subject’s choice in the baseline and the test phase. If the subject walked towards a partner and his/her front paws crossed the line that was dug in the ground 2 m inside the test enclosure (Fig 1) and his/her head was oriented towards a partner, this was considered as a choice. If the subject did not approach either partner within one minute, it was considered a “no-choice” response.

“Fig 3. Dogs’ and wolves’ attentiveness in the control and experimental condition” Please add if the behavior was towards both partners and during the observational phase.

We have created a new figure (Fig 3, see below), which combines the two graphs in the previous version to make things clearer, and changed the caption (line #492):

Fig 3. Dogs’ and wolves’ attentiveness towards the generous and selfish partner in the observation phase in the control and experimental condition. The black diamond depicts the mean.

We also changed the description in the main text (line #485):

Specifically, dogs and wolves were significantly more attentive towards both partners in the control condition than the experimental condition (p = .044). Further, the interaction between species × partner shows that dogs were equally attentive to the generous and selfish partner but wolves were significantly more attentive to the generous partner than the selfish one in both conditions (p = .039) (Fig 3, Table 2).

Moreover, the differences between the generous and the selfish person were subtle, with few ostensive cues, making it very difficult to distinguish between them.

Although this is possible, we argue that the difference between the generous and the selfish partners’ actions were not too subtle. We have addressed this point in the discussion (line #564):

A possible explanation why the animals did not form reputations of the humans is that they did not distinguish between the generous and the selfish partners’ actions. This could account for the negative results in Jim et al. [27], where both partners performed the same pushing action but the outcome was different. However, in the current study, the differences between the partners’ actions were greater – the selfish partner crossed her arms and turned her back from the dog demonstrator, whereas the generous partner did not turn her back from the demonstrator and threw the piece of meat into the enclosure. Furthermore, the dog demonstrator’s reactions to the partners were different too; the dog ate the food after interacting with the generous partner and did not eat after interacting with the selfish partner.

In contrast with the generous person, the selfish one kept the food in her hands. The longer duration of the presence of the food associated with the selfish human could increase the attentiveness and the preference towards her. So, the human attitude could compete with the presence of the food, overshadowing a clear preference for one person. You must include this limitation in the discussion.

This is a good point and we have included it in the discussion (line #573):

The differences in the partners’ actions are important for the animals to be able to discriminate between them, but the fact that the selfish partner kept the food in her hand may have been a potential confound; the longer duration of the presence of the food was associated with the selfish partner and may have led some animals to choose her. Therefore, the selfish partner’s actions may have competed with the presence of food and overshadowed a clear preference for one partner, which may explain our non-significant results. It would have been better to have the selfish partner act ‘competitively’ and eat the food in front of the animal like in Heberlein et al.’s studies [16, 37], but this was not possible in this study because we used pieces of raw meat as rewards to increase the animals’ motivation, thus the selfish partner could not eat it.

I think that the hypothesis regarding the importance of more experience (more demonstration trials) cannot be discarded. On the one hand, many works have shown that the performance of the animals improves with additional training trials, especially in difficult tasks like this. On the other hand, the fact that some animals showed a preference for the generous person could mean that they were faster discriminating both human attitudes. Therefore, the ability can be present in these species but some animals need more experience to achieve their goal.

I agree that it seems logical that performance of animals improves with additional training trials, but surprisingly, I could not find any empirical studies to support this statement. If the reviewer can provide some references, I will gladly add it to the discussion. We also recently conducted a similar study to this one, where we increased the number of demonstrations/experiences in the observation/experience phase, but the results were still non-significant. We have included this unpublished study in the discussion (line #592):

It is possible that we may have found evidence for indirect and direct reputation formation in wolves and dogs if they had observed more demonstrations/experiences in the observation/experience phase, respectively, and more trials in the test phase. Therefore, we replicated the current study with pet dogs and increased the number of demonstrations/experiences from four to six per partner in the observation/experience phase and the number of trials from six to twelve in the test phase, but we still found no evidence of neither indirect nor direct reputation formation (unpublished).

It is fascinating that wolves are more attentive than dogs in several tasks. This difference could be related to different feeding ecologies. However, the explanation is not clear to me. Even when dogs do not depend on conspecifics to obtain food, they strongly depend on humans to do that. So, it would be expected for them to be very attentive and looked at humans during the demonstrations longer than wolves. In this experiment it is more important to pay attention to the human than to the dog demonstrator.

I agree and we have included this in the discussion (line #627):

One could speculate that dogs should pay more attention to the humans than wolves because they rely on humans for valuable resources, such as food and shelter [25], but studies have shown that wolves pay more attention when observing others’ behaviours or to details compared to dogs. For example, Range and Virányi [31] found that wolves can follow human and conspecific gaze and can even follow a human gaze cue into distant space, while dogs did not [32]).

Additionally, we have included a possible explanation for why wolves were equally attentive to the partners as dogs were in line #639:

As the wolves at the WSC are hand-raised by humans, it is possible that they are equally attentive to humans and conspecifics; 

In my opinion, one important factor is the low level of getting attention cues the animals received during the entire task. The authors did not report the duration of the attentive response of the animals during the observation phase, so it is hard to know which level of attention they showed. Considering that this is a complex task, the time animals spent looking at the demonstrations is crucial.

We were unable to code the duration animals spent looking at the demonstrations from the video footage. We explain why in detail in the discussion (line #643): 

The animals at the WSC are used to seeing familiar humans (the trainers) interact with conspecifics in situations involving food, such as during shifting or training, and they are used to seeing unfamiliar humans interact with conspecifics in contexts without food during weekly pack visits. Unfamiliar humans are not allowed to have food around the WSC animals, thus we thought they would be more attentive to the unusual situation that the unfamiliar partners had pieces of meat that they could feed to them. That being said, our experimental setup still may not have been interesting enough for the animals. It is crucial for the subject to pay attention to the partners’ actions to understand their different roles to form reputations of them. We coded the animals’ attentiveness as a binary variable (i.e., whether the subject’s head was oriented towards the direction of the interaction) since we could not code attentiveness as a continuous variable (i.e., duration) for two reasons: first, we could not place the camera closer to the subject by placing it inside the test enclosure because it may have obstructed the subject’s view of the third-party interactions. Second, the observer’s area was quite spacious (e.g., 19 m2) to allow the wolves to move around freely, as tight spaces are stressful for them. However, this meant that the animal was sometimes out of view or too far away to see whether they were paying attention to the interaction.

How frequently did you have to interrupt the evaluation due to the animals’ lack of motivation? Was it similar for dogs and wolves?

We did not interrupt the evaluation (i.e., test phase) if they were not motivated. We only stopped the test phase if the animal was stressed, and this only happened with one wolf (Taima) in the first condition.

I think it is very difficult to measure an animal’s lack of motivation in the test phase and different people could have a different operational definition for this. For example, it could be defined as the subject not making a choice consecutively in over half of the trials (i.e., at least 4 out of 6 trials) in sessions 3 and 4. As you can see in S2 Dataset, most of the animals made at least 3 choices in a session. The only exceptions are: two dogs, Hiari (not motivated in sessions 3 and 4 in the first condition) and Enzi (not motivated session 3 in the first condition), and one wolf, Geronimo (not motivated in session 4 in the second condition). Even so, we continued with all the trials in the test phase even if the animal was not motivated, as each trial was only for one minute and there was a total of six trials in one session. Moreover, these individuals only lacked motivation in one of the conditions, therefore I would argue that all the animals were relatively equally motivated to participate in the study overall.

In addition, it is possible that for dogs, the interaction between the human and the other dog evokes other emotions like jealousy. You must include this in the discussion.

This is an interesting point but I doubt the interaction between the human and the dog demonstrator would have evoked emotions like jealousy in our study. Harris and Prouvost (2014) studied jealousy in dogs – they defined “the proposed function of jealousy is to break-up a potentially threatening liaison and protect the primary relationship” and found that dogs exhibited jealous behaviours towards their owner. In our study, the subjects had never met the two unfamiliar partners before, thus they did not have a relationship to be jealous. Apart from this, we would expect that the wolves would be equally jealous since they also show inequity aversion as dogs (Essler et al., 2017). Also, if the animals were jealous, it is unclear what the prediction would have been – perhaps the animals would have chosen the generous partner more often because they wanted to be close to her, but we did not find this.

---

## [Decision Letter · Decision Letter 1]

4 Jul 2022

Wolves and dogs fail to form reputations of humans after indirect and direct experience in a food-giving situation

PONE-D-21-40254R1

Dear Dr. Jim,

We’re pleased to inform you that your manuscript has been judged scientifically suitable for publication and will be formally accepted for publication once it meets all outstanding technical requirements.

Kind regards,

Christoph Englert

Academic Editor

PLOS ONE

Additional Editor Comments (optional):

Reviewers' comments:

Reviewer's Responses to Questions

**Comments to the Author**

1. If the authors have adequately addressed your comments raised in a previous round of review and you feel that this manuscript is now acceptable for publication, you may indicate that here to bypass the “Comments to the Author” section, enter your conflict of interest statement in the “Confidential to Editor” section, and submit your "Accept" recommendation.

Reviewer #1: All comments have been addressed

2. Is the manuscript technically sound, and do the data support the conclusions?

Reviewer #1: (No Response)

3. Has the statistical analysis been performed appropriately and rigorously? 

Reviewer #1: (No Response)

4. Have the authors made all data underlying the findings in their manuscript fully available?

Reviewer #1: (No Response)

5. Is the manuscript presented in an intelligible fashion and written in standard English?

Reviewer #1: (No Response)

6. Review Comments to the Author

Reviewer #1: I appreciate the efforts of the authors to address all of my previous comments. In my opinion, the manuscript is now suitable for publication.

7. PLOS authors have the option to publish the peer review history of their article (what does this mean?). If published, this will include your full peer review and any attached files.

Reviewer #1: No

---

## [Editor Report · Acceptance letter]

22 Jul 2022

PONE-D-21-40254R1 

Wolves and dogs fail to form reputations of humans after indirect and direct experience in a food-giving situation 

Dear Dr. Jim:

I'm pleased to inform you that your manuscript has been deemed suitable for publication in PLOS ONE. Congratulations! Your manuscript is now with our production department. 

Kind regards, 

on behalf of

Dr. Christoph Englert 

Academic Editor

PLOS ONE